human-computer interaction/applied mathematics

ransomware, price discrimination, ransom, willingness to pay, cyber-security

**Author for correspondence:**
E. Cartwright
e-mail: edward.cartwright@dmu.ac.uk

# An economic analysis of ransomware and its welfare consequences

J. Hernandez-Castro[1], A. Cartwright[2] and E. Cartwright[3]

[1]School of Computing, University of Kent, Canterbury, Kent, UK
[2]School of Economics, Finance and Accounting, Coventry University, Coventry, UK
[3]Department of Economics and Marketing, De Montfort University, Leicester, UK

EC, 0000-0003-0194-9368

We present in this work an economic analysis of ransomware, a relatively new form of cyber-enabled extortion. We look at how the illegal gains of the criminals will depend on the strategies they use, examining uniform pricing and price discrimination. We also explore the welfare costs to society of such strategies. In addition, we present the results of a pilot survey which demonstrate proof of concept in evaluating the costs of ransomware attacks. We discuss at each stage whether the different strategies we analyse have been encountered already in existing malware, and the likelihood of them being implemented in the future. We hope this work will provide some useful insights for predicting how ransomware may evolve in the future.

## 1. Introduction

Ransomware refers to the branch of malware that, after infecting a computer, asks for a ransom. The term can be used generally to denote all kinds of extortion. For instance, it includes malware that puts compromising or illegal material on to a computer and then asks for a ransom not to report the victim to the police. In our work, we are interested in a more subtle kind of malware, originally called cryptovirus but later also referred to as crypto-ransomware or simply ransomware. In this case, the malware encrypts and then deletes the original data files and asks for a ransom to hand them back to the victim [1,2].

The original concept of cryptovirus was first presented in the academic literature by Adam Young and Moti Yung around 1996 [3]. It was probably inspired by previous unsuccessful attempts to extort money out of infected computers by, among others, the AIDS malware. The key development in the Young and Yung approach was to employ public-key cryptography for performing this extortion in a cryptographically sound and robust manner. Robust here means that the scheme is not vulnerable to key compromise by reverse-engineering, as so frequently occurred

with substandard malware using a symmetric key [4]. In other words, a victim has no choice if they want to recover their files other than to interact with the criminals (and potentially pay the ransom) in order to recover the relevant key.

For some time, the concept of cryptovirus remained well known by academics but was only scarcely used by malware developers, who apparently stumbled upon it and fine-tuned it independently. The case of the Gpcode ransomware is particularly interesting in this regard, in that we can see a criminal experimenting and making multiple mistakes and numerous failed attempts before 'getting it right in the end' [5]. Cryptolocker, discovered in the wild in 2013, was one of the first, if not the first, to implement a scheme close to the Young and Yung protocol in a technically sound way, from its conception [6]. Cryptolocker demonstrated the potential to extract large amounts of money through a cryptovirus (see below). Since then, there has been an explosion in the number of different ransomware families and variants found (CryptoWall, TorLocker, Fusob, Cerber and TeslaCrypt to name a few) building up an industry estimated to be worth up to $1 billion a year [1,7–9].

The crucial point that motivates the current paper is that we would expect the criminals are refining their techniques, not only in terms of the malware component technology, but also regarding the *economic tools* they use to extract money from victims. In this regard, Cryptolocker was relatively unsophisticated when compared with some modern strands. Just as the criminals learned by trial and error how to launch a cryptovirus, it is natural that they will also learn better strategies to profit from their crime. In this paper, we look at some of the strategies criminals could use to extract illegal gains from ransomware. Topics that we cover include the optimal ransom and the potential for price discrimination. We conjecture that future attacks will probably evolve, slowly but surely, toward an optimal economic strategy, and so our paper offers a view on what the future of ransomware may look like.

Importantly, we also address the policy and welfare implications of ransomware. In particular, we look at how to measure the welfare costs of ransomware and how these costs may change if ransomware develops as we predict. Crucially, we shall demonstrate that our model gives critical insight on: (i) the strategies criminals can employ, and probably will employ, to profit from ransomware, together with (ii) the welfare costs to society of such strategies. For instance, we show that the criminals have an incentive to increase the ransom they charge and focus on the victims who most value their files. If they were to do this, it would significantly increase the welfare costs of ransomware. The model is already being used as a tool for law enforcement to measure the potential threat of emerging ransomware strands [10].

To calibrate our theory, we report the results of a small-scale survey (conducted on students in the UK) on willingness to pay (WTP) and accept (WTA) for loss of files. The survey is not designed to give definitive results but to demonstrate how the welfare costs of ransomware can be estimated. We find that the welfare cost of ransomware using current strategies is estimated to be around £150 per individual. If the criminals were to 'improve' their strategies the welfare costs would rise to around £250. Moreover, the profits of the criminals would approximately double. Individual losses of £250 may sound relatively manageable; for instance, it compares to average losses from credit card fraud (in the UK) of over £800 per individual [11] and car theft of over £4000 [12]. But ransomware attacks are getting easier to launch and that means a large number of people are potentially going to fall victim. If just 1% of the UK adult population were to be attacked, the estimated losses are £131million.[1] These numbers, provisional though they are, illustrate that ransomware is a serious and growing threat to society. And there is no reason to suspect that the situation in the UK is any different from other developed countries.

The rest of the paper proceeds as follows. In §2, we provide some basic background on ransomware. In §3, we discuss the demand function for files. In §4, we look at uniform pricing and in §5 at price discrimination. We conclude in §6.

## 2. Background on ransomware

Before we focus on the economics of ransomware let us briefly provide some background information. The basic idea behind ransomware is very simple in that malware encrypts files on a computer and asks for a ransom. As already mentioned, if done in a sound way, the only way to recover the files is to pay the ransom and receive the relevant key. Victims are given a set time, typically 72 h, to pay the ransom, which can vary from $100 to $1000 for individuals, and a lot more for firms and organizations [8]. It is worth highlighting that modern ransomware is run as a business operation with 'customer-service' to facilitate paying of the ransom [13].

---

[1]Based on a UK adult population of around 50 million people.

Ransomware presents a straightforward crime of financial extortion. Historical examples date back to, at least, the AIDS Trojan Horse released in 1989 which demanded a $189 ransom from its victims. It is only in the last 10 or so years, however, that we have seen ransomware become a common and developing threat. This owes much to the difficulty of producing a technically sound cryptovirus. Many criminals still commit the error of using private key cryptography, which can be reverse-engineered by skilled professionals [4]. But even if criminals were familiar with the academic work on cryptovirus, the proper development and implementation of such a complex piece of software is still fraught with multiple opportunities to incur fatal technical errors.

We have seen examples where the means for recovering the key or decrypting the files were quickly found and shared by the security community, leading to negligible business for the cyber-criminals. A good example of this flawed ransomware is TorLocker, active in 2014 and first targeted at Japanese users, in which despite all files being encrypted with AES-256 and RSA-2048, in more than 70% of the cases, the key can be obtained and the files decrypted due to errors made during the implementation of cryptography algorithms [14]. But there are many other examples, such as Cryptear, Hidden-Tear, EDA2, Stampado, NoobCrypt, Crypt0, 777, Petya, KeRanger, CryptoDefense, earlier versions of Torrentlocker and Jigsaw where, due to programming errors, there was little if any revenue for the criminals [15].

Cryptolocker was the first of the modern strand of cryptovirus, and one of the most costly so far, in part due to its good implementation and security. Its 'good' implementation unfortunately forced victims wanting to recover their files to pay the ransom. Survey estimates suggest that anything up to 40% of victims paid the ransom and around two-thirds of those recovered their files after paying the ransom [16]. While the precise proportion of victims who paid ransoms is unknown, it is clear that enough people paid the ransom to generate a large amount of money. Conservative estimates on the amount of ransom received by the criminals range from $300 000 to over $1 000 000 [17–19]. We also know that a single Bitcoin address connected with cryptolocker (174psvzt77NgEC373xSZWm9gYXqz4sTJjn) received a total of 346 102 BTC at the time of its last transaction in February 2014. This was a significant proportion of the total number of Bitcoins in circulation (approx. 12 million) and would have had a valuation in excess of $200 m.

Having connected Cryptolocker with Bitcoin, it is interesting to highlight that Bitcoin and other crypto-currencies have played a fundamental role in the 'success' of Cryptolocker and other recent ransomware (although we should highlight that other means of payment have been used such as MoneyPak) [4]. Indeed, the development of crypto-currencies is arguably as important to the emergence of ransomware as the technology knowledge to write a cyptovirus. Bitcoin allows for relatively easy money transfers and, although by no means untraceable or completely anonymous, it is considered to be secure enough to provide a good degree of anonymity. These characteristics provide cyber-criminals with a very powerful tool for profiting from their crimes, and one that law enforcement is not that accustomed to. They can use Bitcoin to defeat the classical control measures already put in place to trace, follow and stop other, better-known payment methods such as bank transfers.[2]

Operation Tovar in 2014, led by the US Department of Justice and the FBI but also involving Europol, law enforcement from several countries, multiple security firms and some universities, led to the Gameover/Zeus botnet being closed down. This was one of the main distribution paths for Cryptolocker and since its take-down this particular cryptovirus can be considered to be eradicated. During Operation Tovar, a victims database was located, containing approximately 500 000 individuals, and this allowed the set-up of a website to facilitate victims recovering their files. Note, however, that this was only possible due to the recovery of the criminals' database and not to any security weakness in the implementation of the cryptovirus itself. This initiative has been followed by the No More Ransom Project that aims to help victims recover files where possible (https://www. nomoreransom.org/).

Naturally, after the widely reported 'success' of Cryptolocker, there has been an explosion in the number of ransomware strands [21,22]. There are now readily available development kits selling in the black market alongside Ransomware-as-a-Service (RaaS). Until the only known solution to these threats (i.e. offline backups) is widespread, ransomware is going to be an attractive proposition to criminals. Indeed, all the evidence suggests that ransomware is on the increase and will become an ever more

---

[2]Despite its clear advantages for criminal activity, Bitcoin also has disadvantages in terms of extracting money [20]. There may well be a significant number of victims willing to pay a ransom but that lack the ability to access Bitcoin, particularly in short time periods as requested by some ransomware. This friction is easing over time as crypto-currencies become more popular. A further difficulty with Bitcoin-based ransoms is the characteristic volatility of Bitcoin. This sometimes has forced criminals to adjust the ransom request, priced in Bitcoins, to keep it in line with the real cost in dollars, sterling or euros [17].

common threat in the years to come. And it should be recognized that while high-profile attacks such as the WannaCry virus in May 2017 attract a lot of headlines, ransomware is a daily threat with new and effective variants such as Cerber, GandCrab and Katyusha emerging all the while [23].

Ransomware has become a modern threat because of the technical advances in cryptovirus and crypto-currency. Overcoming the technical challenges of mounting a malware attack is, though, as we have already said, only part of the story. If the criminals want to make money then their economic strategy is going to be crucial. We, therefore, naturally expect that the economic strategy of the criminals is likely to evolve as they become more experienced with ransomware. It is essential that we are ahead of the game in terms of predicting future development for, at least, two reasons. First, to foresee the likely welfare costs of ransomware for society in order that we can act accordingly by, say, prioritizing advice to individuals and firms. Second, so that we can reliably predict which ransomware strands are likely to be a higher threat in terms of welfare costs to society and illegal gains for the criminals. Indeed, the evidence suggests that a handful of ransomware strands have been responsible for most of the damage [19]. An explosion in the number of ransomware strands in the wild means it becomes ever more important for law enforcement and others that we can pinpoint which strands are likely to 'survive and flourish'.

# 3. The demand function for files

Ransomware is, by its nature, a financially motivated crime. We will, therefore, assume that the objective of the cyber-criminals is to maximize profit.[3] Given that it is relatively costless to attack victims and there is little chance of being caught, the main variable the criminals can control is the size of ransom. This will, therefore, be the focus of our analysis. Note that in taking this approach, our analysis is best suited to randomly distributed attacks, such as Cryptolocker and, more recently, Ryuk, that are designed to infect as many computers as possible, and where the criminal has very limited information about victims before the attack. In this setting, the criminals maximize profit, as we shall see shortly, by setting a ransom based on 'average' WTP a ransom across the population. Our approach is less well suited to precisely targeted attacks in which a particular organization or firm is targeted for extortion (such as a university or health trust). In this setting, the criminals are likely to have more precise information on the ransom the victim may be willing to pay.

At the point where a criminal has infected a computer, we have a game with two players—the criminal who holds the key to the files and the victim who would like to recover his or her files. We shall analyse this game using models of monopoly pricing in which the criminal is viewed as the unique 'seller' of a product, namely the key to decrypt the files, that the victim would be willing to buy. The models that we shall use are very standard in economics [24], but they are not so well known outside of economics and so we shall provide sufficient detail here for all to follow. Crucially, we shall demonstrate that the models give critical insight on: (i) the strategies criminals can employ, and probably will employ, to profit from ransomware, as well as (ii) the costs to society of such strategies.

The profit that criminals can make largely depends on the willingness of those attacked to pay the ransom. This, in turn, will depend on various components—how important the files are to the victim, whether they have a recent backup, how much liquid money they have available, the extent to which they trust the criminals to honour their word, willingness to give money to criminals, etc. From the criminal's perspective, however, it is irrelevant why people are, or are not, willing to pay a ransom. All that matters is the maximum amount a particular victim is *willing to pay* to recover their files.[4] Different people will naturally have a different WTP, and so we denote by $v_i$ the WTP of person $i$. For instance, a victim who values her files at \$500, has no recent backup, and trusts the criminals will return her files would have $v_i = 500$ while a victim who values her files at \$1000 but has a recent backup, or does not trust the criminals and dislikes interacting with criminals may have $v_i = 0$.

The revenue of the criminals can then be summarized as

$$\Pi = \sum_{i=1}^{N} (p_i - c)1_i - F,$$

---

[3]We use the word profit in its economic sense. This is not to imply that the money generated from ransomware is legitimized in any way.

[4]Exogenous, random shocks that result in a victim not being able to pay (because, say, they cannot access Bitcoin) are also irrelevant from the criminal's perspective.

where $N$ is the number of people attacked, $p_i$ is the ransom asked of person $i$, $c$ is the cost of dealing with any ransom money and providing a service to victims, $1_i$ is an indicator variable that takes value 1 if $p_i \leq v_i$ and 0 otherwise, and $F$ is the fixed cost of operating the malware. In the following, we shall abstract away from considering $N$ and simply take it as given that the criminals will target as many people as possible.[5] Our focus will, thus, be on deriving the optimal ransom to charge victims. Note that we also abstract away from dimensions other than money, such as the amount of time and effort a victim would be willing to devote to dealing with the criminals, obtaining Bitcoin, etc.

The welfare loss to victims from the attack can be bounded from below by

$$W = \sum_{i=1}^{N} (p_i 1_i + v_i(1 - 1_i)).$$

This expression recognizes that those who pay the ransom lose the ransom payment (but potentially regain their files) and those who do not pay the ransom lose their files. This expression is a lower bound because victims who pay the ransom may not regain their files. Moreover, those who are not willing to pay a ransom to criminals may still, as discussed above, value their files highly. In other words, WTP $v_i$ may not capture the full value of the files to the victim. So, the welfare loss may be higher than $W$. Even so, we shall see that $W$ provides a meaningful (and estimable) lower bound on the welfare consequences of ransomware.

We see that the optimal strategy of the criminals and the welfare consequences of ransomware will depend critically on the WTP of victims. The relevant information on the WTP can be encapsulated in a 'demand function for the criminal's services'. Specifically, let $Q(p)$ denote the proportion of victims willing to pay a ransom of value $p$. In the following, we derive the optimal ransom demand and welfare consequences as a function of $Q$. To add extra insight we complement the theoretical discussion with some survey estimates of $Q$ derived from university students. Let us clarify upfront that we make no claims that our sample is representative of the general population. We shall see, however, that our survey demonstrates proof of concept and allows us to calibrate the economic theory in an informative way.

In order to obtain a demand function based on genuine data, we conducted a face-to-face survey using standard contingent valuation techniques [26]. The survey was performed at the University of Kent (UK) over three sessions. All sessions took place in a classroom at the university and were widely advertised around the campus. One session was run on a University Celebration day and so primarily involved alumni and local residents while the two other sessions primarily involved students. A total of 149 respondents took part (54% male, average age of 24). The survey was the second part of a five-part questionnaire that participants completed under experimental conditions (i.e. informed consent and full anonymity).[6] Participants were paid based on their choices in the final part of the questionnaire and received an average payment of around £6. The overall questionnaire took around 20 min to complete.

In moving to our survey data, it is important to clarify that $Q$ is the demand function of *victims*. In other words, it summarizes the WTP of people who fall victim to a ransomware attack (because they are prone to phishing, do not have anti-malware, etc.). We implicitly take it as given that anybody in the population is a potential victim of ransomware and so the general population is representative of victims. We, therefore, focus on eliciting WTP rather than measuring a person's defence against attack. This seems an appropriate first approximation given that people consistently overestimate their ability to prevent cyber-attack [27,28].

Half of those surveyed were asked the following two questions:

1. Suppose that because of a mistake you made, you have lost access to all of the files on your computer. The only way you can recover the files is to pay a private company who are experts in file recovery. What is the maximum amount you would pay to recover your files?
2. Suppose that your computer was infected by a virus which means you cannot access any of your files. The criminals responsible have been caught and you are now eligible for monetary compensation. How much money would you want to recompense you for the loss of files? Note that if your request is deemed too high the authorities will use a technique to recover your files and so you will receive your files but no compensation.

---

[5]For more on the strategies criminals may use to target different types of victim see [25].

[6]The other parts of the questionnaire are unrelated and not discussed here.

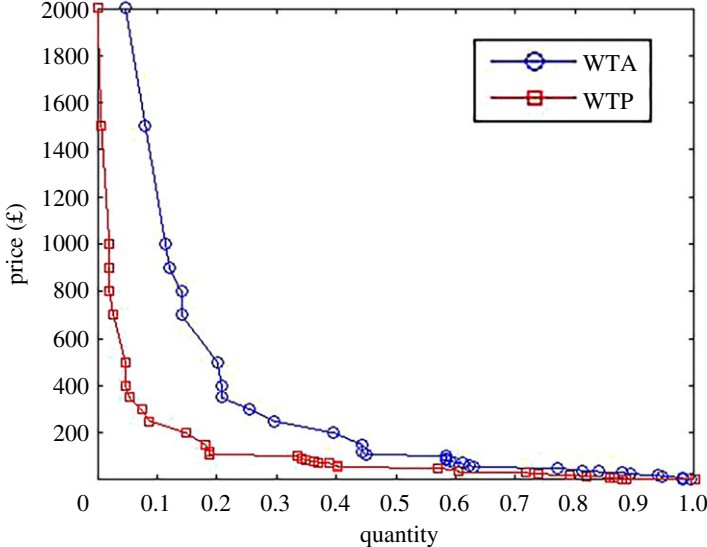

**Figure 1.** Demand curve elicited using WTA and WTP.

Question 1 directly elicits WTP—how much is someone willing to pay to recover their files. We know, however, that people have a tendency to understate WTP. Question 2 addresses this problem by eliciting WTA—how much would someone need to be paid in order to compensate for the loss of files. Note that to make a WTA question incentive-compatible one needs to invoke a default mechanism for payment to stop subjects over-stating the value of loss. In our case, we invoke the authorities. Provided that subjects do not feel they can predict how the authorities would value their files, it is in the interests of subjects to state their true WTA.

In principle, WTP and WTA should be identical. Typically, however, one obtains a WTA–WTP disparity in that WTA is significantly higher than WTP [29]. This difference can be anything from a factor of 2 to 10. The true valuation $v_i$ can reasonably be assumed to lie between an individuals' stated WTA and WTP. However, there are arguments to suggest that the true valuation will be closer to WTA than to WTP [30].

As a robustness check, the other half of those surveyed were asked the following questions:

3. Suppose that because of a mistake you made, you cannot access any of your files. You have an insurance policy that means you are eligible for monetary compensation. How much money would you want to recompense you for the loss of files? Note that if your request is deemed too high, a technique will be used to recover your files meaning you will receive your files but no compensation.
4. Suppose that your computer was infected by a virus which means you cannot access any of your files. The only way you can recover your files is if you pay a fee to the criminals. If you can be certain that your files will be returned, what is the maximum you would pay to recover your files?

Note that question 3 elicits WTA while question 4 elicits WTP. Reversing the order of questions allows us to check that respondents' stated WTA and WTP is not influenced by the order in which the questions are asked. Moreover, the framing varies between questions 1 and 4, and 2 and 3 in terms of who was responsible for the loss of files and who will be paid or recompensed.

Ideally, we would want answers to questions 1 and 4 to be similar, and those to questions 2 and 3 to be similar. This was indeed the case. Specifically, we performed the non-parametric Wilcoxon rank-sum test to check that answers to questions 1 and 4 are from the same distribution ($p = 0.55$) and that the answers to questions 2 and 3 are from the same distribution ($p = 0.22$). So, in the following, we shall solely report WTP (questions 1 and 4) and WTA (questions 2 and 3).

It is conventional to plot the inverse demand curve $Q^{-1}(p)$. Figure 1 plots the inverse demand curve using the elicited values of WTA and WTP with total demand (149 subjects) normalized to 1. Note that price is in sterling, and at the time of the survey, the exchange was around \$1.40 per £1.00. As expected, the WTA (mean £719, median £100, standard deviation £4178) exceeds the WTP (mean £103, median £50, standard deviation £189) resulting in higher demand when using the WTA as compared to the WTP. For instance, 20% of those surveyed were willing to accept at least £400 and pay up to £100. This is a

WTA–WTP disparity of factor 4. A related survey by IBM found that 54% of people would pay up to $100 to recover their files [31]. This estimate (figure 1) fits within our WTA and WTP, although is closer to our WTA.

# 4. Uniform pricing

In order to maximize profit, the criminals must determine an optimal ransom to ask each victim. In an ideal world, they would know the WTP $v_i$ of each victim and ask a ransom just below $v_i$. This is clearly not representative of current ransomware. In this section, we consider the more reasonable assumption that the criminals rather know the distribution of $v_i$'s within the population. In other words, they know the demand function $Q$ but do not know where any particular individual's valuation lies on the demand function. In justifying an assumption that $Q$ is, we note that ransomware criminals have access to abundant data, from previous victims, on willingness to pay a ransom. This means they should be able to form a reasonably accurate belief about the demand function $Q$ [32]. Moreover, as we discuss below, it is not necessary for the criminals to know $Q$ in order to act as if they do know $Q$.

In the case where the criminal knows only the demand function $Q$, it is optimal to use uniform pricing. This means the criminal sets the ransom to the same amount for anyone attacked, $p_i = p$ for all $i$ [24]. This appears to be the approach currently taken by most criminals and ransomware strands.

In the case of uniform pricing, the profit of the criminals can be written as

$$\Pi = (p - c)NQ(p) - F,$$

where, we recall, $Q(p)$ is the proportion of people willing to pay a ransom of value $p$. Maximizing profit with respect to $p$ (and assuming $Q$ is differentiable), we get the optimal ransom demand $p^\star$ given by

$$\frac{p^\star - c}{p^\star} = -\frac{1}{\eta(p^\star)},$$

where $\eta(p) = (p/Q(p))\mathrm{d}Q(p)/\mathrm{d}p$ is the price elasticity of demand. Price elasticity measures the sensitivity of demand to changes in the ransom. It is optimal for the criminals to price where demand is elastic, i.e. $|\eta(p^\star)| \geq 1$, meaning that an increase in the amount of the ransom would lead to a more than proportionate fall in the number of people paying.

To illustrate, we can work out the optimal ransom demand for our estimated demand function. One approach is to simply search for the ransom that maximizes revenue. In our case, this is £1500. To provide an approach less sensitive to individual subjects, we fitted a (six-degree) polynomial to the raw demand function elicited using WTA.[7] From this fitted demand function, we can then calculate marginal revenue (MR) using equation

$$\mathrm{MR}(Q) = \frac{\mathrm{d}p(Q)}{\mathrm{d}Q}Q + p(Q).$$

The optimal ransom demand is found where MR equals marginal cost. It seems reasonable to assume that the marginal cost to the criminals of dealing with an additional victim is near zero. The optimal ransom is, therefore, found by setting $\mathrm{MR}(Q) = 0$.

Figure 2 plots the raw demand function (the same as in figure 1), the fitted demand function and MR. Note that there are five values of $Q$ for which $\mathrm{MR}(Q) = 0$. Only one of these is the global optimum, and it is simple to show that this is the smallest $Q$ where $\mathrm{MR}(Q) = 0$. This gives an optimal ransom of around $p^\star = $£950. It is predicted that, in this setting, around 10% of victims will pay (and 90% will not). An interpretation of this finding is that it is in the criminal's interest to target high-value victims who are willing to pay £1000 or above. For instance, at the optimum value, the expected profit of the criminals is £99 per victim (because a little over 10% of victims will pay £950). If the ransom is dropped to, say, £150 then over 40% of victims will pay but this only results in a profit per victim of around £60.

One implication of optimal pricing illustrated above, which many find counterintuitive, is that (in the case of uniform pricing) the criminals should set the ransom at a level where a sizable proportion of victims *will not be willing to pay the ransom*. For our fitted demand function, the proportion was 90%. More generally, if $Q(p)$ is convex to the origin, then less than 50% of the victims should be willing to pay the ransom if it is set optimally. Recall that survey estimates of the proportion of victims who pay

---

[7]The fitted equation is $p = -37\,950Q^5 + 116\,699Q^4 - 137\,561Q^3 + 77\,678Q^2 - 21\,367Q + 2472.1$. This is fitted to minimize the sum of squared differences between the fitted and predicted values.

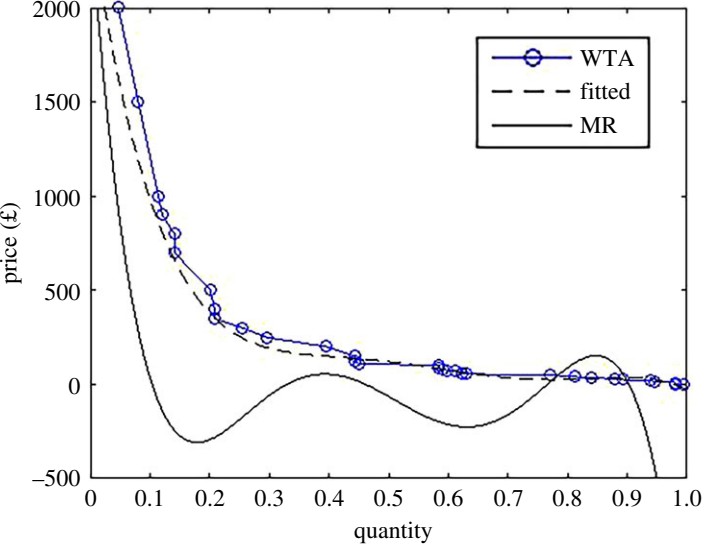

**Figure 2.** Demand curve elicited using WTA and marginal revenue (MR).

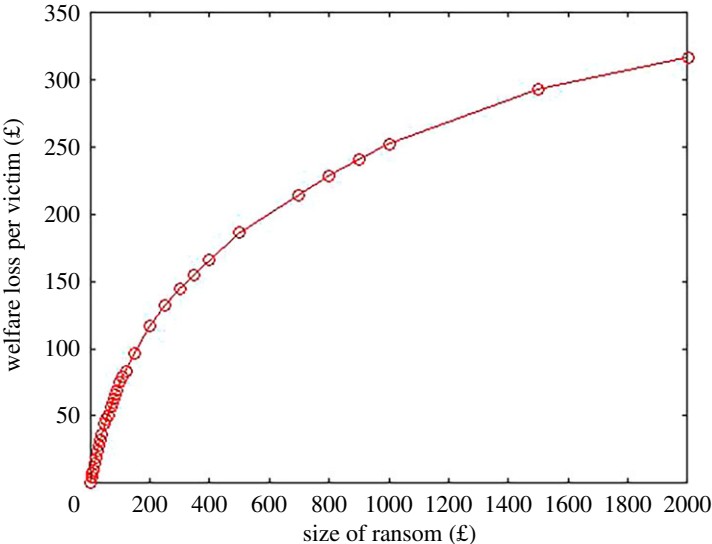

**Figure 3.** Predicted welfare loss for survey data.

a ransom range from 2 to 40%. We can see that interpreting these numbers is necessarily going to be difficult because it will be hard to distinguish a 'failed' attack where fewer people pay than the criminals expected versus an attack that was 'successful' but in which few paid the ransom. For instance, in our setting recall that the optimal ransom means only 10% pay the ransom.

This distinction can be crucial when it comes to welfare analysis. The cost to society of an attack can be bounded from below by

$$W = pNQ(p) + N \int_0^p Q(p)p.$$

This expression recognizes that proportion $Q(p)$ pays the ransom and the rest lose their files. Figure 3 plots the welfare loss from various ransom amounts from our survey data. You can see that the welfare loss is increasing in the size of ransom. It is trivial to see that this is a general property because a higher ransom means a higher proportion will not pay (and therefore lose their files) and those who do pay are paying a larger amount. If, therefore, it is optimal for the criminals to set a relatively high ransom in which few victims pay, we can see that the welfare loss is likely to be high. For instance, at a ransom demand of £950, the predicted welfare loss is around £250 per victim on average. If the ransom demand were lower at £300 (the kind of demand we saw with Cryptolocker) then the welfare loss would be reduced to £140 per victim.

In most realistic scenarios, the criminals do not know the *a priori* distribution of $v_i$'s and so do not know $Q(p)$. But they can hypothesize that, under very mild and general assumptions, the value of price elasticity $|\eta(p)|$ is a weakly decreasing function of $p$. This makes it relatively easy for the criminals to heuristically find out whether the current ransom they are using is set too high or too low. For example, suppose the current ransom is \$300 and the criminals observe that 40% of people pay the ransom, $Q(300) = 0.4N$. The criminals could try increasing the ransom to \$350. Suppose that at this higher price around 35% of victims pay, $Q(350) = 0.35N$. Then, the price elasticity can be approximated by $\eta(300) = -(300/0.4N)(0.05N/50) = -0.75$. If the marginal cost of attack $c$ is approximately zero, then, at the current ransom price, demand is too inelastic. It would, in this case, be optimal to increase the ransom above \$300.

By systematically varying the ransom over time, the criminals can learn the elasticity of demand and quickly converge on the optimal price [32]. Interestingly, fluctuations in Bitcoin value provide a natural experiment on the sensitivity of demand to price (in dollars). For instance, the ransom demanded by Cryptolocker changed from around 2 BTC to 1 BTC, 0.5 BTC and then 0.3 BTC over the course of a few months in late 2013 [17]. This was presumably done to keep the ransom amount stable in dollar terms. It would, however, provide an indirect means to estimate the elasticity of demand. It seems highly unlikely the criminals exploited this, or similar, possibilities. Instead, it seems ransoms are set relatively haphazardly, although there has been an upward trend in the ransom demand over time [33]. Given the incentives to determine an optimal ransom, we would not be surprised to find schemes employed in the near future that actively experiment with price and (deliberately or not) search for information about demand, as given by $Q(p)$, and set the ransom accordingly.

# 5. Price discrimination

In the previous section, we assumed that criminals could not distinguish between different victim's WTP. This is a reasonable assumption, supported by many examples of ransomware strands in the wild, but it limits the profits criminals can make to that of a uniform pricing monopolist. They can certainly increase their profit by price discriminating, that is, by setting a different ransom for different victims. In the following, we explore ways in which to do that. Let us begin by noting the common distinction between three types of price discrimination [34].[8] First degree (or perfect) price discrimination involves each person being given an individually tailored price based on their WTP. In other words, a victim with the WTP of $v_i$ is given a ransom just below $v_i$. This is the 'gold standard' of discrimination that criminals may ultimately achieve but seems a long way off at the moment. More realistically criminals can use second- or third-degree price discrimination.

Third-degree price discrimination is simplest to apply and so we shall focus on that in the remainder of the paper. Before doing that let us briefly comment on second-degree price discrimination. This would see victims offered a menu of packages. For instance, there might be a basic package of \$100 to recover word processing files, a better package of \$200 that includes personal photos, and a complete package of \$300 that includes all files. This would seem very easy for criminals to do (from a technical point of view), but to the best of our knowledge, there is no ransomware that has employed anything of this nature. The objective in offering a menu of options is to screen victims so as to extract additional profit. For instance, current ransomware only offers an 'all-inclusive' package. You may have a victim who is not willing to pay a high ransom but would be willing to pay a smaller ransom to recover, say, some word documents.

Third-degree price discrimination offers a slightly more direct way of distinguishing different types of victim. Access to a person's files and computer can give the criminal's useful information about the value of the information present on the computer, the victim's worth, or both. For example, they can look at the number, type and size of files, model and age of the computer, etc., and search for sensitive or valuable data. Any or all of this information may correlate with a person's WTP. If so, the criminals can categorize victims into types and apply third-degree price discrimination. Indeed, criminals with a large base of victims can apply machine learning, regression or clustering techniques to a series of easily extracted features in a victim's system.

To illustrate how this may work with a very simple example, suppose the criminals can distinguish people with a 'large' number of files from those with a 'small' number of files. They could then price-discriminate by setting a ransom $p_L$ for anyone with a large number of files, and $p_S$ to anyone with a

---

[8]The terms first-, second- and third-degree price discrimination are an accident of history and do not convey any substantive meaning other than being a classification used by economists.

small number of files. For each type of victim, the optimal price is determined as explained above for uniform pricing. So, the optimal ransom for those with a large number of files is

$$\frac{p_L - c}{p_L} = -\frac{1}{\eta_L(p_L)},$$

where $\eta_L$ is the price elasticity of demand for people with a large number of files. Similarly, the optimal ransom for those with a small number of files can be determined from the analogous elasticity $\eta_S$. We would expect those with a larger number of files to have, on average, a higher WTP to recover their files. This would imply that, for any given price, demand is less elastic for those with a larger number of files, $|\eta_S(p_S)| > |\eta_L(p_L)|$. Thus, the criminals should charge a higher price to those with a large number of files, $p_L > p_S$.

Charging two different prices increases the criminals' profit by making the ransom more personalized. But there is no reason other than simplicity to have only two different prices. The more information the criminals can extract on the expected WTP the more personalized should be the ransom. By systematically varying the ransom, and by taking into account the extracted information about victims, the criminals can learn over time the optimal ransom for each 'type' of victim. The more personalized the pricing becomes, the closer the criminals come to attaining the maximum achievable profit, corresponding to first-degree price discrimination [34]. One reason price discrimination increases profits is because it increases the number of people paying the ransom. This works because the criminals can lower the ransom for those with a lower WTP, *without lowering* the ransom for those with a higher WTP. We shall discuss the welfare consequences of this shortly. First, let us briefly comment on the practical ways in which ransomware may manifest itself in the wild.

It is interesting that some ransomware families started to implement some sort of primitive third-degree price discrimination strategies. For example, the Shade strand is using a Remote Access Trojan (RAT) to spy on their victims finances (apparently a version of TeamViewer) before estimating the ransom they can request [35]. This is a poor way to implement price discrimination, because it is time-consuming, not easily scalable and opens the criminals to potentially being traced back more easily. But, it shows the potential for price discrimination. As another example, in the latest variant of Fantom, the ransom amount is determined by the name of the process/executable file, thus allowing for multiple simultaneous attack campaigns, targeting different types of victims and requesting different ransom amounts [36]. Possibly the cleanest example of price discrimination is GandCrab which adjusts the ransom demand based on the type of files available and the size of the database [37].

Moreover, a simple and popular attack vector for ransomware is malvertising, which consists of malicious ads that can be delivered by popular and reputable websites (such as AOL, The New York Times, BBC, MSN, etc.). These ads will (typically by using an iframe) quietly install an exploit kit and, as part of it, a ransomware. Well-known examples of ransomware families using this strategy are the Cerber 3.0, Locky and TeslaCrypt. This is popular between cyber-criminals because it does not require user intervention, can reach potentially millions of users (through Google and other major advertising networks) and can cost as little as 30 cents per 1K impressions [38]. Malvertising would seem particularly relevant for price discrimination because criminals can target customers with a certain profile, for example, those belonging to a given demographic or those who have searched for certain keywords. Thus, basically abusing the ad network services as a proxy for cheap customer classification. No evidence of malvertising for price discrimination has been documented so far. But it does provide a dangerous threat that we should be aware of.

To explore the consequences of price discrimination for welfare let us return to our survey data. Recall that at the optimal ransom of £950 only around 10% of victims would pay. Given that 90% of victims will not pay such a high ransom it is clearly in the criminal's interest to price discriminate in order to get more victims to pay. Ideally, the criminals would want to identify those willing to pay a high ransom and ask a high ransom while making a lower ransom demand to others. This, though, requires being able to discern those willing to pay a high ransom. This could be based on data in the infected computer or the personal characteristics of the victim. For instance, we find in our data that the WTP and accept is on average lower for women (WTP £84, WTA £505) than men (£119, £898), although the differences are not significant ($p > 0.25$ Mann–Whitney test). We also find the WTP and accept is increasing with age (regression coefficient 0.004 for WTP and 0.016 for WTA) where the effect is statistically significant for WTA ($p < 0.05$).[9]

---

[9]We ran ordinary least-square regressions with log(WTP+1) and log(WTA+1) as the dependent variables and gender and age as independent variables. The age variable is not significant for WTP ($p = 0.55$) but is for WTA ($p = 0.02$).

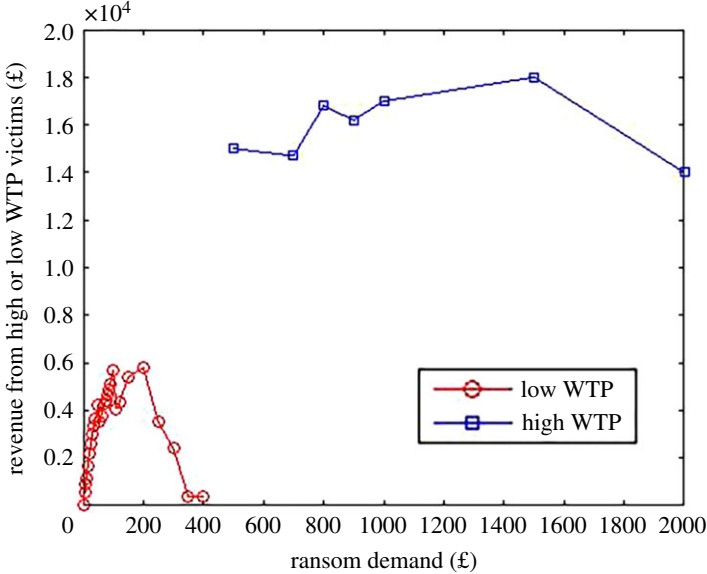

**Figure 4.** Predicted revenue for low and high valuation victims.

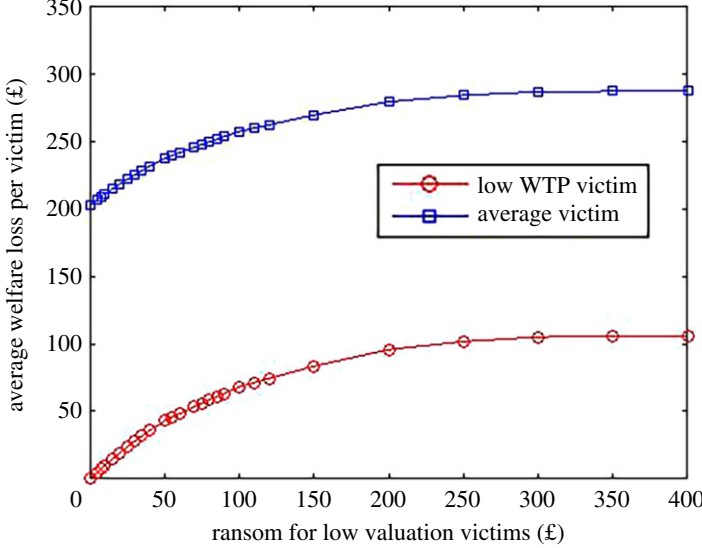

**Figure 5.** Predicted welfare loss for low WTP victims and average (high or low WTP) victims. Based on a ransom of £950 for high WTP victims.

So suppose that the criminals could devise some test that determines, for each victim $i$, whether, say, $v_i > 500$. In other words, they are able to segment victims into those with a high and low WTP. Figure 4 starkly illustrates the consequences of this by plotting revenue for different ransom demands (based on survey values of WTA), where low and high valuation customers are given different ransom demands. For low valuation customers, the optimal ransom is around £100 to £200. But you can see that this generates relatively little revenue compared with that from high valuation victims. In particular, high-value victims, even though they are only 20% of our participants, are where the main revenue source is. For instance, at a high ransom demand of £950 there are 17 of our participants predicted to pay the ransom giving revenue of £16 150; at a low ransom demand of £100, there are 57 participants predicted to pay the ransom but this only generates a revenue of £5700.

We can see in this stylized example the trade-offs to price discrimination. On the one hand, price discrimination allows the criminal to generate more revenue. Essentially all of the revenue made from low demand customers is additional revenue compared to what would be made with uniform pricing. So, he would increase revenue by around 33%. This is clearly a significant gain. But, on the

other hand, high valuation customers are the main revenue source and so it would be no surprise to see criminals focus on this group. In short, the main tendency will probably be toward extracting more revenue from the high valuation customers (essentially optimal uniform pricing) but alongside that there are still significant gains from exploiting low valuation customers (through price discrimination).

We turn now to directly look at welfare. Given that price discrimination increases the proportion of victims who pay the ransom (and potentially get their files back), it can *increase* welfare (compared to uniform pricing). The welfare gains, however, are likely to be small. Figure 5 plots the average welfare loss per victim as a function of the ransom for the low valuation victims, on the assumption of a ransom of £950 for high valuation victims. We can see here that the overall welfare loss is largely driven by high valuation victims (even on the assumption they get their files back). To judge the consequences of price discrimination, we need to see what happens to welfare as the ransom for low valuation victims drops from above £400 down to, say, £100. We can see that this does increase welfare because a higher proportion of victims get back their files by paying the ransom. The effect, however, is not dramatic. In particular, the average welfare loss drops by around £30–£50. While price discrimination can decrease welfare loss somewhat it is likely that these gains will be more than offset by the fact the criminals make a larger profit.

# 6. Conclusion

There can be no doubt that ransomware is already a serious security threat to individuals, firms and beyond [39]. It is natural, though, to expect that this threat will evolve over time as the criminals learn from experience and refine their techniques [1]. Indeed, with something in place similar to a selection of the fittest, it is very likely that ransomware will slowly evolve toward criminals using the profit maximizing strategies that can be known in advance thanks to economics and game theory.

In this paper, we have given critical insight on how economics can be applied to study ransomware. A theoretical model of ransomware, based on the standard economic theory of monopoly pricing, was complemented by survey evidence on WTP. Clearly, our survey results can be further developed, for example with a larger and more representative sample, and so, our findings are not definitive. Our results do, however, demonstrate proof of concept and our main findings are qualitatively robust. Moreover, we believe that our findings can be useful for law enforcement in tackling the threat of ransomware. Our findings can be summarized as follows:

— The optimal ransom for attacks on individuals is likely to be larger than we currently see in the wild. A survey suggested that the average ransom demand increased from $294 to $679 between 2014 and 2016 [8]. We would not be surprised to find that ransom demands increase further in the future. A higher ransom means the welfare costs of ransomware will increase, and so ransomware is likely to become more costly to society.
— At the optimal ransom, there will be relatively few victims who pay. This recognizes that the objectives of the criminals are to maximize profit and not the number of people paying the ransom. Evidence that few people pay ransoms should not, therefore, be interpreted as evidence that ransomware is 'not working for the criminals'. We have seen that the main source of revenue for criminals will be high valuation victims, and so it is natural they will target these. This means that the proportion of victims who pay is a poor proxy for the threat of a ransomware strain.
— It is in the interests of the criminals to experiment with ransom demands, or use natural variation in the value of crypto-currencies like Bitcoin, to discern the optimal ransom and indirectly learn more about the WTP of victims. It would be relatively effortless for the criminals to do this (or outsource to someone who can) and so it would seem only a matter of time before it happens. If this does lead to higher ransom demands then it will increase the welfare cost of ransomware.
— The criminals can increase profits through price discrimination. One way is to offer the victim a menu of options at different prices (second-degree price discrimination) while another is to charge different victims a different ransom based on discernible characteristics (third-degree price discrimination). Both forms of price discrimination appear implementable from a technical point of view. We would certainly expect to see more evidence of third-degree price discrimination in the near future. Price discrimination may lower the welfare costs of ransomware by reducing the number of victims who lose their files. But the effects are likely to be small and outweighed by the costs to society of criminals raising more money. Again, the welfare costs to society of ransomware are only likely to increase in the future.

While data on current activity is somewhat limited, we would suggest that the techniques currently being used by the criminals are relatively unsophisticated. There certainly seems to exist ample scope for them to refine their techniques, notably for determining the optimal ransom and to make use of price discrimination. Crucially, refining strategies in the ways we have suggested in this work could lead to dramatic increases in profits at relatively little costs. In particular, the criminals should have at their disposal a wealth of data regarding the willingness of victims to pay a ransom. With only a rudimentary analysis of this data, they could almost certainly obtain higher profits. A more thorough analysis could push profits up even more. With strong incentives for the criminals to innovate, they are surely going to do so.

These findings should be a concern for law enforcement agencies. The profits from ransomware are only likely to increase and so ransomware is not going to go away any time soon. Moreover, as the criminals refine their strategy the welfare costs to society of ransomware will increase. So, how to combat this? We can begin by reiterating that the criminal's optimal strategy and profit is entirely dependent on people's WTP to recover their files. This WTP will naturally depend on some factors that are under the control of the criminals and some that are under the control of the victims. For instance, one thing the criminals control is whether to return files (or, alternatively, keys) to those who pay the ransom. The higher the probability of the files being recovered, the larger will be the WTP, and the larger the ransom the criminals can set [40]. In order to maximize profit, the criminals should, therefore, always return files to people who pay the ransom, unless the cost of returning the files is very large [41].

More important for us is to consider factors under the control of victims. One such factor is the 'value' of the files. If a person has just backed up their files then their WTP is zero. Clearly, this is the easiest way to counteract the criminals. The crucial and non-trivial point to emphasize, though, is that a person backing up her files benefits everyone who might be attacked, and not just herself. This is because the more people back up their files, the lower is demand, the lower becomes the optimal ransom, and the lower are the welfare costs. In other words, a person's decision to back up their files generates a positive externality for all other potential victims.

Another thing under the control of people is whether or not to pay. The fewer people pay the ransom then, again, the lower is demand, the lower the optimal ransom and the lower are the welfare costs. The decision to not pay also, therefore, generates a positive externality. The more a victim internalizes the two externalities we have mentioned, i.e. the more she takes into account the effect that backing up and paying a ransom will have on others, then the lower will be her WTP.

The preceding two paragraphs point to ways in which the impact of ransomware can be diminished. The idea that people need to back up their files is obvious. Less obvious, but potentially powerful, is the role of positive externalities. Strong social norms towards regularly backing up files and not paying a ransom (in the case the files were not backed up) greatly undermine the profit potential of cyber-criminals. While we have focused on individuals here it would seem particularly important that firms, who have shown a ready WTP ransoms, also take these points on board. This may create a remit for policymakers to highlight more strongly the *social* responsibilities around cyber-behaviour. It is also important to consider the incentives for software developers to sell low-cost protection against ransomware [42].

Ethics. Ethical approval was obtained from the School of Economics at the University of Kent. The survey was anonymous and involved informed consent.

Data accessibility. The data from the survey is freely available on FigShare at https://figshare.com/authors/Edward_Cartwright/4438759 [43].

Authors' contributions. J.H.-C. conceived of the study and helped draft the manuscript. E.C. helped draft the manuscript, helped with the survey collection and carried out the statistical analysis. A.C. carried out the survey collection, helped with the statistical analysis and helped draft the manuscript. All authors gave final approval for submission.

Competing interests. We have no competing interests.

Funding. This project has received funding from the European Unions Horizon 2020 research and innovation programme, under grant agreement no. 700326 (RAMSES project). The authors also want to thank EPSRC for project EP/P011772/1, on the EconoMical, PsycHologicAl and Societal Impact of RanSomware (EMPHASIS), which supported this work.

Acknowledgements. We thank Darren Hurley-Smith and other members of the RAMSES project for their feedback on earlier versions of the paper.

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
