## [Reviewer comments · Royal Society Open Science]

Review History

RSOS-190023.R0 (Original submission)

Review form: Reviewer 1

Is the manuscript scientifically sound in its present form?

Yes

Are the interpretations and conclusions justified by the results?

No

Is the language acceptable?

Yes

Is it clear how to access all supporting data?

Yes

Do you have any ethical concerns with this paper?

No

Have you any concerns about statistical analyses in this paper?

Yes

Recommendation?

Major revision is needed (please make suggestions in comments)

Comments to the Author(s)

This paper studies the economics of ransomware, focusing on the victims' willingness to pay and the cybercriminals' pricing strategies. I believe that this is an interesting and important topic since the number and impact of ransomware infections has grown significantly over the past few years. Overall, the paper is well written and easy to follow.

The main contribution of the paper is a survey that studies the participants' willingness to pay for recovering their files. Unfortunately, the survey seems to be using a convenience sample, comprised mostly of university students. I am afraid that this introduces a significant bias since students may significantly differ from the general population in terms of financial situation, age, reliance on their personal computers, etc. Consequently, I have some reservations about the numerical results.

Further, I am concerned about how some of the questions were phrased. In particular, Questions 2 and 3 ask how much compensation would participants request from the government or an insurer after suffering a ransomware attack. The answer here may be influenced by the participants' beliefs about how the government or an insurer would value their files, which is not the same as how they would value their files. Also, the survey results do not account for the participants' ability to make payments (e.g., create a Bitcoin transaction), only for their willingness.

Figure 2 presents a fitted curve. How was this curve fitted? How well does it fit the data? There seem to be significant discrepancies for lower quantities, which may be an issue since the optimum is in this range. By the way, the optimal ransom value could have easily been found for the original data using a simple exhaustive search.

"willingness to pay is on average lower for women (mean £84) than men (mean £119)"
This actually seems like a significant difference. It would have been interesting to learn how willingness depends on certain demographic factors and, even more interestingly, on factors that could be inferred by ransomware and used for price discrimination.

It seems like some parts of the paper were written a while ago. For example, the paper refers to a 2016 survey as "recent" (in the rapidly evolving area of ransomware, 2016 is hardly recent). It would be a good idea to include some more recent references.

Review form: Reviewer 2

Is the manuscript scientifically sound in its present form?

Yes

Are the interpretations and conclusions justified by the results?

Yes

Is the language acceptable?

Yes

Is it clear how to access all supporting data?

Yes

Do you have any ethical concerns with this paper?

No

Have you any concerns about statistical analyses in this paper?

No

Recommendation?

Accept with minor revision (please list in comments)

Comments to the Author(s)

This paper presents an economic analysis of ransomware, a relatively new form of cyber-enabled extortion. The paper is well written with adequate theory, statistical analysis methods and justification of results.

Figure resolutions can be improved for better resolution.

Decision letter (RSOS-190023.R0)

23-Sep-2019

Dear Professor Cartwright,

The editors assigned to your paper ("An economic analysis of ransomware and its welfare consequences") have now received comments from reviewers. We would like you to revise your paper in accordance with the referee and Associate Editor suggestions which can be found below (not including confidential reports to the Editor). Please note this decision does not guarantee eventual acceptance.

Please submit a copy of your revised paper before 16-Oct-2019. Please note that the revision deadline will expire at 00.00am on this date. If we do not hear from you within this time then it will be assumed that the paper has been withdrawn. In exceptional circumstances, extensions may be possible if agreed with the Editorial Office in advance. We do not allow multiple rounds of revision so we urge you to make every effort to fully address all of the comments at this stage. If deemed necessary by the Editors, your manuscript will be sent back to one or more of the original reviewers for assessment. If the original reviewers are not available, we may invite new reviewers.

- Data accessibility

<http://datadryad.org/submit?journalID=RSOS&manu=RSOS-190023>

- Competing interests

- Authors' contributions

- Acknowledgements

- Funding statement

Kind regards,
Andrew Dunn

Royal Society Open Science Editorial Office
Royal Society Open Science
openscience@royalsociety.org

on behalf of Prof Marta Kwiatkowska (Subject Editor)
openscience@royalsociety.org

Associate Editor's comments:

Thank you for your patience while we conducted review of your paper. The journal has had to approach an unusually large number of reviewers to receive these comments. With the difficulty in finding reviewers in mind, we're particularly keen to emphasise that this decision is not a guarantee of eventual publication and that acceptance will be contingent on satisfying the reviewers that your paper is ready for acceptance upon receipt of your revision.

Comments to Author:

Reviewers' Comments to Author:
Reviewer: 1

Comments to the Author(s)

This paper studies the economics of ransomware, focusing on the victims' willingness to pay and the cybercriminals' pricing strategies. I believe that this is an interesting and important topic since the number and impact of ransomware infections has grown significantly over the past few years. Overall, the paper is well written and easy to follow.

The main contribution of the paper is a survey that studies the participants' willingness to pay for recovering their files. Unfortunately, the survey seems to be using a convenience sample, comprised mostly of university students. I am afraid that this introduces a significant bias since students may significantly differ from the general population in terms of financial situation, age, reliance on their personal computers, etc. Consequently, I have some reservations about the numerical results.

Further, I am concerned about how some of the questions were phrased. In particular, Questions 2 and 3 ask how much compensation would participants request from the government or an insurer after suffering a ransomware attack. The answer here may be influenced by the participants' beliefs about how the government or an insurer would value their files, which is not the same as how they would value their files. Also, the survey results do not account for the participants' ability to make payments (e.g., create a Bitcoin transaction), only for their willingness.

Figure 2 presents a fitted curve. How was this curve fitted? How well does it fit the data? There seem to be significant discrepancies for lower quantities, which may be an issue since the optimum is in this range. By the way, the optimal ransom value could have easily been found for the original data using a simple exhaustive search.

"willingness to pay is on average lower for women (mean £84) than men (mean £119)"
This actually seems like a significant difference. It would have been interesting to learn how willingness depends on certain demographic factors and, even more interestingly, on factors that could be inferred by ransomware and used for price discrimination.

It seems like some parts of the paper were written a while ago. For example, the paper refers to a 2016 survey as "recent" (in the rapidly evolving area of ransomware, 2016 is hardly recent). It would be a good idea to include some more recent references.

Reviewer: 2

Comments to the Author(s)

This paper presents an economic analysis of ransomware, a relatively new form of cyber-enabled extortion. The paper is well written with adequate theory, statistical analysis methods and justification of results.

Figure resolutions can be improved for better resolution.

Author's Response to Decision Letter for (RSOS-190023.R0)

See Appendix A.

RSOS-190023.R1 (Revision)

Review form: Reviewer 2

Is the manuscript scientifically sound in its present form?

Yes

Are the interpretations and conclusions justified by the results?

Yes

Is the language acceptable?

Yes

Do you have any ethical concerns with this paper?

No

Have you any concerns about statistical analyses in this paper?

No

Recommendation?

Accept as is

Comments to the Author(s)

The paper is in good shape to be accepted for publication.

Review form: Reviewer 3

Is the manuscript scientifically sound in its present form?

Yes

Are the interpretations and conclusions justified by the results?

Yes

Is the language acceptable?

Yes

Do you have any ethical concerns with this paper?

No

Have you any concerns about statistical analyses in this paper?

No

Recommendation?

Major revision is needed (please make suggestions in comments)

Comments to the Author(s)

I feel that the mathematical depth of this paper is somehow questionable for a journal publication.

However, the insights are useful and they can act as proof of concept. Also, the experiments using students are useful to elicit desired samples.

Some assumptions related to the distribution of v_i can be questionable, e.g. these are known to the attacker - how can the attacker really know this - it seems almost impossible.

I suggest authors include the following:

- 1) the recovery level of a user should they get their data encrypted, e.g. if there is a very recent backup why should they pay the ransom? based on that, different ransom costing will be done.
- 2) the probability of a user being able to pay a ransom. Like this probabilistic analysis can lead to expected returns and then it will need to be translated to some real-world advice.
- 3) It is imperative to justify the selection of the questions asked to students. Why are these the best questions to ask for a problem like this? Why knowing how much they are willing to pay is more important than whether they are prepared in terms of cyber incidents decreasing the probability of the harm?

Decision letter (RSOS-190023.R1)

04-Dec-2019

Dear Professor Cartwright:

Manuscript ID RSOS-190023.R1 entitled "An economic analysis of ransomware and its welfare consequences" which you submitted to Royal Society Open Science, has been reviewed. The comments of the reviewer(s) are included at the bottom of this letter.

Please submit a copy of your revised paper before 27-Dec-2019. Please note that the revision deadline will expire at 00.00am on this date. If we do not hear from you within this time then it will be assumed that the paper has been withdrawn. In exceptional circumstances, extensions may be possible if agreed with the Editorial Office in advance. We do not allow multiple rounds of revision so we urge you to make every effort to fully address all of the comments at this stage. If deemed necessary by the Editors, your manuscript will be sent back to one or more of the original reviewers for assessment. If the original reviewers are not available we may invite new reviewers.

To revise your manuscript, log into <http://mc.manuscriptcentral.com/rsos> and enter your

Author Centre, where you will find your manuscript title listed under "Manuscripts with Decisions." Under "Actions," click on "Create a Revision." Your manuscript number has been appended to denote a revision. Revise your manuscript and upload a new version through your Author Centre.

- Ethics statement

- Data accessibility

- Competing interests

- Authors' contributions

- Acknowledgements

- Funding statement

on behalf of the Associate Editor, and Professor Marta Kwiatkowska (Subject Editor)
openscience@royalsociety.org

Associate Editor Comments to Author:

As one of the original reviewers was not available to assess your revision, we've received comments from a third referee. Given their concerns are relatively substantial, we'd like you to revise the paper to address those concerns. Do bear in mind, however, that no further rounds of revision will be granted.

Reviewer comments to Author:

Reviewer: 2
Comments to the Author(s)

The paper is in good shape to be accepted for publication.

Reviewer: 3
Comments to the Author(s)

I feel that the mathematical depth of this paper is somehow questionable for a journal publication.

However, the insights are useful and they can act as proof of concept. Also, the experiments using students are useful to elicit desired samples.

Some assumptions related to the distribution of v_i can be questionable, e.g. these are known to the attacker - how can the attacker really know this - it seems almost impossible.

I suggest authors include the following:

- 1) the recovery level of a user should they get their data encrypted, e.g. if there is a very recent backup why should they pay the ransom? based on that, different ransom costing will be done.
- 2) the probability of a user being able to pay a ransom. Like this probabilistic analysis can lead to expected returns and then it will need to be translated to some real-world advice.
- 3) It is imperative to justify the selection of the questions asked to students. Why are these the best questions to ask for a problem like this? Why knowing how much they are willing to pay is

more important than whether they are prepared in terms of cyber incidents decreasing the probability of the harm?

Author's Response to Decision Letter for (RSOS-190023.R1)

See Appendix B.

RSOS-190023.R2 (Revision)

Review form: Reviewer 3

Is the manuscript scientifically sound in its present form?

Yes

Are the interpretations and conclusions justified by the results?

Yes

Is the language acceptable?

Yes

Do you have any ethical concerns with this paper?

No

Have you any concerns about statistical analyses in this paper?

No

Recommendation?

Accept as is

Comments to the Author(s)

Contributions seem sufficient and appropriate to be published in your journal

Decision letter (RSOS-190023.R2)

27-Jan-2020

Dear Professor Cartwright,

It is a pleasure to accept your manuscript entitled "An economic analysis of ransomware and its welfare consequences" in its current form for publication in Royal Society Open Science. The comments of the reviewer(s) who reviewed your manuscript are included at the foot of this letter.

on behalf of Prof Marta Kwiatkowska (Subject Editor)
openscience@royalsociety.org

Reviewer comments to Author:

Reviewer: 3

Comments to the Author(s)

Contributions seem sufficient and appropriate to be published in your journal

Appendix A

Reply to Referees on paper 'An economic analysis of ransomware and its welfare consequences' for Royal Society Open Science.

We would like to thank the associate editor and two reviewers for looking at the paper and their constructive feedback. In our understanding, both reviewers were overall positive about the paper and what we were trying to do. Reviewer 1 makes a number of useful comments and suggestions which we have hopefully addressed in the paper and this reply. Reviewer 2 suggested we upgrade the quality of figures, which, again, we have tried to do.

In the following we provide detailed replies to each of the points made. At this stage we note that the main changes in the paper are to better clarify certain points, such as the role and design of the survey.

Before we get into the details let us briefly take a step back to clarify why we feel the paper makes a strong contribution to the literature. When we started working on this topic (around 2015) there were no papers on the economics of ransomware. So, this was a new topic. Over time, papers have emerged, indeed some building on an earlier working version of the current paper (which has 17 citations already on Google Scholar). Our paper is, to the best of our knowledge, the only paper in this literature that sets out a general model to predict the evolution of ransomware demands and quantify the welfare consequences. The model is being used by law enforcement (through the RAMSES project).

Reply to Reviewer 1

This paper studies the economics of ransomware, focusing on the victims' willingness to pay and the cybercriminals' pricing strategies. I believe that this is an interesting and important topic since the number and impact of ransomware infections has grown significantly over the past few years. Overall, the paper is well written and easy to follow.

Thanks for these positive comments.

The main contribution of the paper is a survey that studies the participants' willingness to pay for recovering their files. Unfortunately, the survey seems to be using a convenience sample, comprised mostly of university students. I am afraid that this introduces a significant bias since students may significantly differ from the general population in terms of financial situation, age, reliance on their personal computers, etc. Consequently, I have some reservations about the numerical results.

Personally, we do not feel that the survey is the main contribution of the paper. We feel the main contribution is the theoretical framework with which to evaluate the welfare effects of ransomware. We would argue that we have some nice insights in this regard. Indeed, the model has been turned into a software tool that is now being used by law enforcement (through the RAMSES project). The survey complements the analysis by showing how the theory can be calibrated to real world

data. In particular, the survey is designed to achieve two objectives: (1) A proof of concept that the model can be calibrated. (2) A test of whether the framing of the question influences responses.

It is true that the results should not be seen as representative of the general population. We recognised that in the previous version (to quote 'Let us clarify up front that we make no claims that our sample is representative of the general population. We shall see, however, that our survey demonstrates proof of concept and allows us to calibrate the economic theory in an informative way.'). Interestingly, however, we have run follow up large scale surveys with the general population of the UK. Those results will be reported elsewhere because our objectives were different - we look in detail at how individual characteristics influence WTP but do not look at the effect of framing. The results with the general population were essentially identical to those reported here.

To bring together both of these points let us highlight one thing the survey demonstrates. If you look at standard theoretical analysis it would be conventional to assume a linear demand function Q . Based on that, one could derive a host of predictions. Our survey data, shows, however, that Q is highly non-linear. That has fundamental implications for the results one derives, such as the proportion of victims who will not pay the ransom. We would strongly argue that the non-linearity in our estimated demand function is not an artefact of a student subject pool. So, calibrating the theory adds value.

To address these points we have made more clearer in the paper the role of the survey, both the motivation behind the survey and the sample population. We would also draw attention to the important theoretical contributions, as summarised in the conclusion.

Further, I am concerned about how some of the questions were phrased. In particular, Questions 2 and 3 ask how much compensation would participants request from the government or an insurer after suffering a ransomware attack. The answer here may be influenced by the participants' beliefs about how the government or an insurer would value their files, which is not the same as how they would value their files. Also, the survey results do not account for the participants ability to make payments (e.g., create a Bitcoin transaction), only for their willingness.

This concern addresses one objective of the survey, namely to check the consequences of different framing. The WTP question is relatively easy to set out – even if there are differences between payment to a private company (question 1) and criminal (question 4). The WTA questions (questions 2 and 3) are necessarily harder to frame because you have to invoke some mechanism to stop subjects asking for a huge amount of money. We followed standard techniques in contingent valuation to frame the questions (in our case with an authority or insurer). It is difficult to see what else we could have done that would have significantly changed this framing. The main alternative would be to say that a value is randomly drawn

from some distribution, but that does not seem a very compelling narrative for subjects particularly as we need to specify the probability distribution etc. We now comment on this in the paper.

The point about Bitcoin is an interesting one in that it recognises the role of effort and time as well as money in recovering files. We comment on this in the paper. Ultimately, we feel that money is likely to be the prime factor in willingness to engage with criminals.

Figure 2 presents a fitted curve. How was this curve fitted? How well does it fit the data? There seem to be significant discrepancies for lower quantities, which may be an issue since the optimum is in this range. By the way, the optimal ransom value could have easily been found for the original data using a simple exhaustive search.

This is a standard polynomial fit to minimize the squared differences between real and fitted values. It is easy enough to calculate the optimal ransom for our raw data is £1500. For the fitted curve we get estimate £950. The optimal ransom will be sensitive to the distribution of those with high valuations and so our numbers are definitely no more than suggestive. Again, however, this does not preclude insight: most ransomware demands from spam distribution are below £300. Our analysis suggests this is less than would be optimal.

"willingness to pay is on average lower for women (mean £84) than men (mean £119)" This actually seems like a significant difference. It would have been interesting to learn how willingness depends on certain demographic factors and, even more interestingly, on factors that could be inferred by ransomware and used for price discrimination.

We now report the non-parametric test result for male versus female. Both, for the non-parametric test and regression analysis we find no statistically significant difference. In subsequent work we are looking in more depth at factors that influence WTP. Ultimately, though, the best indicators are likely to be on the computer itself – number of files, how recently the files were edited etc.

It seems like some parts of the paper were written a while ago. For example, the paper refers to a 2016 survey as "recent" (in the rapidly evolving area of ransomware, 2016 is hardly recent). It would be a good idea to include some more recent references.

For that particularly issue, i.e. changes in the ransom criminals are demanding, we are not aware of any update since the 2016 survey (beyond complete anecdote). In our opinion this is an issue that warrants more study. That said, we have added some fresher academic references from the last few years to bring the literature more up to date.

Reply to Reviewer 2

This paper presents an economic analysis of ransomware, a relatively new form of cyber-enabled extortion. The paper is well written with adequate theory, statistical analysis methods and justification of results.

Thank you for the positive comments

Figure resolutions can be improved for better resolution.

We have redone all the figures in Matlab. Not sure if they comes through in the revised submission, but ultimately means we have the figures in high resolution.

Appendix B

Reply to additional reviewer: An economic analysis of ransomware and its welfare consequences, 2nd round of revisions

We would like to thank the Editor and reviewers for their comments on the revised version of the manuscript.

To the best of our understanding both reviewers 1 and 2 have now expressed a view that the paper is suitable for publication. Our comments here, therefore, deal with the feedback from the new reviewer 3. The additional reviewer has made some useful suggestions which we have acted on in this latest version. Below we detail the changes made; we also indicate these changes as bold font in the main paper. Given the positive response from the 2 previous reviewers our changes are exclusively focussed on capturing the views of the new reviewer.

Detailed reply to Reviewer 3

I feel that the mathematical depth of this paper is somehow questionable for a journal publication. However, the insights are useful and they can act as proof of concept. Also, the experiments using students are useful to elicit desired samples.

We thank you for recognising that the insights are useful. It is true that the mathematics in the paper is a relatively straightforward application of monopoly pricing theory. We would, though, argue that the application is an important one and our theory gives critical insight on ransomware.

Some assumptions related to the distribution of v_i can be questionable, e.g. these are known to the attacker - how can the attacker really know this - it seems almost impossible.

Clearly an assumption that the criminals know the exact distribution of values is extreme. But, we do not believe it unreasonable that the criminals could form beliefs on valuations that are reasonably accurate. The criminals have access to abundant data from previous victims, i.e. the ransom and whether the victim paid. The evidence also suggests that ransomware is increasingly the domain of organised criminals who have the capabilities to learn from this data. We comment on this more in the revised version of the paper. Moreover, our paper is about how ransomware may develop in the future. Our argument would be that the criminals have an incentive to analyse data in order to learn the distribution of valuations. Indeed, Caulfield, Ioannidis and Pym (2018) discuss how to do this. We would also point to page 10 where we discuss the possibility of the criminals converging on the optimal ransom without knowing Q .

I suggest authors include the following:

1) the recovery level of a user should they get their data encrypted, e.g. if there is a very recent backup why should they pay the ransom? based on that, different ransom costing will be done.

The term v_i captures all relevant factors that determine willingness to pay. Whether a person has a recent/partial back-up is a crucial part of the mix. We now explicitly discuss the role of the back-up in the formation of v_i . As we discussed in the conclusion the easiest way to tackle ransomware backups – in which case the v_i 's converge on zero and so ransomware

has no profitable future.

2) the probability of a user being able to pay a ransom. Like this probabilistic analysis can lead to expected returns and then it will need to be translated to some real-world advice.

To be honest we are not entirely sure what the suggestion is here. One thing that is important is the ability of the victim to pay the ransom, i.e. get money in time. So, we have now included this in the list of factors that influence the willingness to pay. But we are not convinced of the need to adopt a probabilistic approach. For instance, suppose there is some exogenous shock that means a victim is not able to pay the ransom (because they cannot get hold of Bitcoin etc.). This has no effect on the optimal ransom from the criminal's perspective. All that matters from the criminal's perspective is the Q function.

3) It is imperative to justify the selection of the questions asked to students. Why are these the best questions to ask for a problem like this? Why knowing how much they are willing to pay is more important than whether they are prepared in terms of cyber incidents decreasing the probability of the harm?

This is an important point that we have now clarified in Section 3. The demand function Q is about victims and so we implicitly assume that the overall population are representative of victims. We consider this valid given that everyone is capable of falling for a ransomware attack. The relevant variable for the analysis is, therefore, WTP and so that is why we focus on this in the analysis. In related work we are exploring how WTP correlates with the cyber-behaviour of individuals and while there is an effect, it is very small in magnitude. Some elements of preparedness, e.g. whether the person has a back-up, are included within WTP. It is also very difficult to measure preparedness given that most individuals have a very poor understanding of the realities of cyber-security.